# Acceptability of the COVID-19 Vaccine and Its Determinants among University Students in Saudi Arabia: A Cross-Sectional Study

**DOI:** 10.3390/vaccines9090943

**Published:** 2021-08-25

**Authors:** Mohammed J. Almalki, Amani A. Alotaibi, Salman H. Alabdali, Ayman A. Zaalah, Mohsen W. Maghfuri, Naif H. Qirati, Yahya M. Jandali, Sami M. Almalki

**Affiliations:** 1Department of Health Services Management, Faculty of Public Health & Tropical Medicine, Jazan University, Jazan 45142, Saudi Arabia; 2Department of Health Education and Promotion, Faculty of Public Health & Tropical Medicine, Jazan University, Jazan 45142, Saudi Arabia; alotaibi@jazanu.edu.sa; 3Department of Epidemiology, Faculty of Public Health & Tropical Medicine, Jazan University, Jazan 45142, Saudi Arabia; 201802961@stu.jazanu.edu.sa (S.H.A.); 201807853@stu.jazanu.edu.sa (A.A.Z.); 201704650@stu.jazanu.edu.sa (M.W.M.); 201701885@stu.jazanu.edu.sa (N.H.Q.); 201807881@stu.jazanu.edu.sa (Y.M.J.); 4Department of Health Informatics, Faculty of Public Health & Tropical Medicine, Jazan University, Jazan 45142, Saudi Arabia; 201803002@stu.jazanu.edu.sa

**Keywords:** acceptability of vaccine, vaccination hesitancy, COVID-19, university students, Saudi Arabia

## Abstract

COVID-19 vaccine hesitancy is a significant threat to the efforts that have been taken to combat the pandemic. This study assessed the acceptability of the COVID-19 vaccine among university students in Saudi Arabia. A cross-sectional online survey using a Google Form was conducted between 2 April and 23 April 2021. A snowball technique was used to recruit participants for this study. The final sample consisted of 407 participants. More than one-third of the participants (36.1%) had received the COVID-19 vaccine, and 13.3% had registered to receive the vaccine. Of the participants who were not yet vaccinated (n = 260), 90.4% indicated that they would like to be vaccinated when given the opportunity. Of the unvaccinated participants, 82.3% trusted the COVID-19 vaccines that had been provided in Saudi Arabia. The most reported reasons for the participants’ acceptance to receive the COVID-19 vaccine included preventive purposes (95.8%), a belief in the safety of the vaccines (84.3%), and the availability of public awareness information regarding the vaccines (77.3%). A small portion of participants (6.1%) were refusing to receive the vaccine due to the potential long-term side effects (92.0%) and expedited vaccine trials (80.0%). Acceptability of the COVID-19 vaccine was strongly associated with participants who regularly received the flu vaccine (*p* < 0.05). All other demographic variables were not statistically associated with the acceptability of the COVID-19 vaccine. In conclusion, it would be appropriate for universities to launch peer programs to urge reluctant students to receive the vaccine voluntarily. In terms of further research, it is valuable to follow up with unvaccinated participants to investigate if they received the vaccine since the data were collected, and their reasons for doing so. This research would reveal changes toward vaccine acceptability over time and any related determinants. Future research should consider students from non-Arabic speaking backgrounds.

## 1. Introduction

Since the emergence of COVID-19 in late 2019 in Wuhan, China, efforts have been undertaken to develop safe and effective vaccines against this disease. As of 29 June 2021, about 14 COVID-19 vaccines were approved by the World Health Organization (WHO) for emergency use. Additionally, there were 105 vaccines in the clinical development stage and 184 vaccines in the pre-clinical development evaluation stage [1]. Furthermore, in line with global efforts and in addition to the other preventive measures, Saudi Arabia launched a free vaccination campaign against COVID-19 for all citizens and residents on 15 December 2020 [2].

The Pfizer/BioNTech and Oxford/AstraZeneca vaccines were reviewed and approved by the Saudi Food and Drug Authority to be used during the vaccination campaign [3]. Registration for the COVID-19 vaccine was provided via the “Sehaty” application. Sehaty is an Arabic word that means “my health”. Sehaty is an application to provide health services to individuals in Saudi Arabia. It allows the user to access health information and obtain a number of online health services provided by various health authorities [4]. The vaccination campaign was carried out in multiple stages [2]. The first stage targeted the elderly (>65 years), obese individuals with >40 body mass index (BMI), patients with two or more chronic diseases or those with immune deficiency, and professionals who were most vulnerable to infection. The second stage targeted citizens and residents over 50 years old and any health practitioners who were not vaccinated in the first stage. It also targeted obese people with a BMI between 30–40 and patients with a single chronic disease. The final stage targeted all citizens and residents of Saudi Arabia.

Vaccine hesitancy and refusal are of major threats to public health [5]. Such behaviors could limit the achievement of vaccination goals [5], and undermine efforts to combat COVID-19 [6]. Vaccine hesitancy is defined as a “delay in acceptance or refusal of vaccination despite [the] availability of vaccination services” [7]. For the purpose of this study, vaccine acceptability is defined as “the acceptance of vaccination”; vaccine hesitancy is defined as a “delay in acceptance of vaccination despite the availability of vaccination services; and, vaccine refusal is defined as “the refusing of vaccination despite the availability of vaccination services”. Globally, evidence showed that many people were hesitant or refused to accept the COVID-19 vaccines [8]. A survey conducted in the United States examined 592 participants’ intentions to receive the COVID-19 vaccine and found that 40.9% were hesitant to obtain the vaccine [9]. Another community-based survey conducted in Portugal indicated that COVID-19 vaccine hesitancy was high, with up to 56.0% of participants indicating that they would delay the vaccine, while 9.0% would refuse it [10]. At the regional level in the Middle East, a survey study in Kuwait found that 53.1% of participants were accepting the COVID-19 vaccination, and 46.9% were definitely or probably hesitant to receive the vaccine [11]. The most recent large-scale study was conducted to measure COVID-19 vaccine hesitancy among 36,220 Arab participants both within and outside of the Arab countries [12]. Results indicated a significant vaccine hesitancy rate, with 81.0% for participants in the Arab countries to 83.0% for Arabs in other countries that were hesitant to receive the vaccine.

In Saudi Arabia, several studies have measured the acceptance level of the COVID-19 vaccine. A study was conducted in early 2020 to measure the intent of the general public to receive a hypothetical COVID-19 vaccine. Results showed that 64.7% of participants intended to accept the hypothetical vaccine, while the rest were hesitant [13]. The hesitancy to receive the COVID-19 vaccination was not limited to the general public in Saudi Arabia. Another survey study found that vaccine hesitancy was highly prevalent among healthcare workers as only 50.5% accepted to receive the COVID-19 vaccine. Recent statistics by the WHO revealed that more than 17 million doses of the COVID-19 vaccine were administered (49.81 doses per 100 population) in Saudi Arabia as of 5 July 2021 [14]. These statistics can shed light on changes in people’s attitudes and acceptance of the COVID-19 vaccine over time.

Similar to the general public and professional groups, university students were hesitant to accept the COVID-19 vaccine. Sallam et al. [15] examined vaccine hesitancy among university students in Jordan. The proportion of participants who accept to receive COVID-19 vaccines was low (34.9%) compared to those who were hesitant (25.5%) or refusing (39.6%) [15]. Another cross-sectional study was performed among Egyptian medical students. Even though they were medical students, 46.0% had vaccination hesitancy, 6.0% definitely refused the vaccine, 13.0% refused with some hesitancy, 29.0% accepted with some hesitancy, and only 6.0% definitely accepted the COVID-19 vaccine [16]. A very recent study from India assessed vaccine hesitancy among 1068 medical students in India from 2 February to 7 March 2021, and found that vaccine hesitancy was only 10.6% [17]. This recent finding may reflect a positive change in the acceptability of the COVID-19 vaccine in students.

The most cited reasons for the COVID-19 vaccine’s hesitancy and refusal included concerns about the vaccine’s safety and efficacy, worries about unknown side effects, expedited vaccine trials, conspiracy theories, and the anti-vaccine movement. Other reasons included lack of eligibility information, refusing the vaccination on principle, the belief that natural exposure to infections provides the safest protection, a lack of trust in government agencies, and personal demographics such as gender and age [8,9,10,11,12,13,14,15,16,17,18,19,20].

Evaluation studies are required to assess the acceptance of the COVID-19 vaccine among university students in Saudi Arabia. Our study aimed to investigate the acceptability of the COVID-19 vaccine among university students in Saudi Arabia and identify any related determinants. 

## 2. Materials and Methods

### 2.1. Study Design, Population and Sample

Our study used a cross-sectional design. The targeted population consisted of Jazan University students in Saudi Arabia, aged 18 years or older, and enrolled in one of the offered courses during the second semester of 2021. Using Epi Info version 7.2.4 [21], the ideal sample size was estimated based on the population size (42,007) [22], a 50% response rate, a 5% margin of error, and a 95% confidence interval, which was calculated to be 384 participants.

### 2.2. Questionnaire

A questionnaire was developed for this study, and it consisted of three parts: Demographics, questions about the acceptability of the COVID-19 vaccine, and an open-ended space for any additional information (See Appendix A). Demographic variables included age, gender, marital status, the discipline of their study, education level, whether the participant had contracted COVID-19, whether any of the participant’s relatives or friends had acquired COVID-19, whether any of the participant’s relatives or friends died from COVID-19, whether the participant regularly received the seasonal flu vaccine, and whether the participant was on the waiting list for the COVID-19 vaccine. The dichotomous variables with yes/no answers were coded 0 for no and 1 for yes. The demographic variables were coded as follows: Gender (0 for male and 1 for female); marital status (0 for not-married and 1 for married); and study discipline (0 for non-health and 1 for health disciplines). Finally, categorical variables with three answers were coded 1, 2, and 3, respectively.

The second part of the questionnaire (acceptability of COVID-19 vaccine) consisted of five main questions on the participants’ trust toward COVID-19 vaccines that have been used in Saudi Arabia, factors influencing the participants’ trust toward COVID-19 vaccines, willingness to receive the COVID-19 vaccine, and factors for their acceptability or hesitancy toward the COVID-19 vaccine. All questions in this part used a five-point Likert scale, with the exception of the question on their willingness to receive the COVID-19 vaccine. The Likert responses ranged from strongly agree to strongly disagree, and coded from 5 to 1, respectively. The question on the participants’ willingness to receive the COVID-19 vaccine had a yes/no option, coded 1 for yes and 0 for no.

The questionnaire was initially developed in English, then translated into Arabic by the first author. Other co-authors, who were fluent in Arabic and English, confirmed the validity of the translation and the integrity of the content. The questionnaire was also presented to the Faculty’s Research Unit and discussed by a public health expert, who confirmed its clarity and relevance. The questionnaire was subsequently tested on a group of university students to ensure its clarity before distribution. Finally, the Arabic version of the questionnaire was distributed to the target population.

### 2.3. Data Collection

An online survey using a Google form was conducted between 2 April and 23 April 2021. A snowball technique was used to recruit study participants, as the researchers distributed the survey link to their WhatsApp contacts and their Twitter followers. The survey link was also distributed through internal emails and students’ WhatsApp groups. The researchers also urged participants to share the survey link with their WhatsApp contacts and followers on social media platforms.

Ethical clearance was obtained from the Research Ethics Committee of Jazan University, Jazan, Saudi Arabia (REC42/1/146). Although participation was anonymous, all participants were asked for their consent before accessing the questionnaire items through a yes/no question regarding their willingness to voluntarily participate in the study. In addition, participants were provided with all required information about the targeted population, instructions for completing the questionnaire, and confidentiality measures applied to protect the collected data.

### 2.4. Data Analysis

IBM SPSS Statistics software version 27 for Windows was used to clean, code, manage, and analyze the data of this study. Frequencies and descriptive statistics were calculated and reported for demographic and background variables, factors of trust toward COVID-19 vaccines, and intentions toward receiving the COVID-19 vaccine and influencing factors. A chi-square test for independence was performed to explore the relationship between the acceptability of the COVID-19 vaccine as a dichotomous (yes/no) dependent variable and each of the following demographics as independent variables: Gender, age, marital status, the discipline of study, contracted COVID-19, a family member or a friend contracted COVID-19, a family member or a friend died from COVID-19, and regularly received the flu vaccine. We used the value of Yates’ Continuity Correction to report the results for the chi-square tests of the 2 × 2 contingency tables, and the Pearson chi-square value for the rest. The Binary logistic regression analysis was employed to identify demographic variables (as a set) associated with the acceptability of COVID-19 vaccine. The tested independent variables included gender, age, marital status, the discipline of study, contracted COVID-19, a family member or a friend contracted COVID-19, a family member or a friend died from COVID-19, and regularly received the flu vaccine. A *p*-value ≤ 0.05 for all tests was considered statistically significant.

## 3. Results

### 3.1. Demographic Description

A total of 502 participants responded to the survey. However, the final sample consisted of 407 participants, as we excluded 95 cases. Of the excluded cases, 81 were duplicated, probably due to technical issues, five cases disagreed to voluntarily participate in the study, and nine cases presented inaccurate information.

The average age of the participants was 21.62 years (SD = 2.54, range 18–36). Table 1 presents the demographic information of the participants. Approximately half of the participants (55.8%) were male, not-married (92.1%), and from a health-study background (55.8%). Most of the participants (88.9%) did not contract COVID-19, however, 81.3% had a family member or a friend who contracted COVID-19. Notably, 20.1% of the participants had a family member or a friend who died due to COVID-19.

### 3.2. Acceptability of COVID-19 Vaccines

The following figures show that 36.1% (*n* = 147) of the total participants had received the COVID-19 vaccine and that 13.3% (*n* = 54) were registered on waiting lists to receive the vaccine. Of the participants who were not yet vaccinated (*n* = 260), 90.4% indicated that they would like to be vaccinated if given the opportunity. Furthermore, 82.3% (*n* = 214) of unvaccinated participants stated they strongly agreed or agreed when they asked if they trusted the COVID-19 vaccines that had been used in Saudi Arabia (Figure 1, Figure 2 and Figure 3).

Important causes given by the participants for their general trust toward COVID-19 vaccines in Saudi Arabia included their trust in the Saudi Government (96.5%), trust in the Saudi health system (94.3%), and community figures who received the vaccine (85.8%). More information is presented in Table 2.

Figure 4 shows the motivations of participants who received or wanted to receive the COVID-19 vaccine. Most of the participants (95.8%) indicated their willingness to protect themselves and others from COVID-19 as their primary motivation for receiving the vaccine, followed by the security provided by and safety of the vaccines (84.3%) and the availability of public awareness information about COVID-19 vaccines (77.3%).

A small portion of participants (n = 25, 6.1% of the total sample) refused to receive the COVID-19 vaccine. The factors identified for their refusal included worrying about long-term side effects (92.0%) and expedited vaccine trials (80.0%). Figure 5 summarizes the factors that drove participants to decline the COVID-19 vaccines.

A chi-square test for independence indicated a significant association between the “regularly received the flu vaccine” variable and the acceptability of COVID-19 vaccine, *X*^2^ (2, *N* = 407) = 7.30, *p* = 0.004, *phi* = 0.144. All other demographic variables were not associated with the acceptability of the COVID-19 vaccine (Table 3). 

The Binary logistic regression analysis was performed using the enter method to assess the impact of the demographic independent variables (as a set) on the acceptability of the COVID-19 vaccine (Table 4). The full model was statistically significant, *X*^2^ (9, *N* = 407) = 18.68, *p* = 0.028. However, only the variable of “regularly received the flu vaccine” made a unique statistically significant contribution to the model (*p* = 0.007, OR = 0.248). Participants who had regularly received the flu vaccine were four times more likely to accept the COVID-19 vaccine than were those with no regular history of receiving the flu vaccine (Table 4).

## 4. Discussion

Many vaccines were manufactured in response to the COVID-19 pandemic. To date, a number of those vaccines have been approved by WHO and other scientific institutions worldwide, and other vaccines are still in the clinical development or pre-clinical evaluation stages [1]. However, people generally indicate concern about the safety of new vaccines, and thus vaccinations are delayed until the safety of the vaccines is verified [23]. The Saudi health authorities approved and utilized several COVID-19 vaccines, including the Pfizer/BioNTech and Oxford/AstraZeneca vaccines [3]. These vaccines were free of charge for all populations—Saudis and non-Saudis [24]. Our study revealed that 36.1% of the total participants received the vaccine within 1 month of its availability in the Jazan region, and 13.3% were registered on waiting lists to receive the vaccine. However, 50.6% of our participants did not register for the COVID-19 vaccine. Of them, 90.4% were willing to receive the COVID-19 vaccine. Although we collected data in the first month after releasing the public vaccination in the Jazan region, the results showed that the acceptability of the COVID-19 vaccine was high compared to prior studies from Saudi Arabia [13,24]. At the regional level, our results regarding the acceptance to receive the COVID-19 vaccine were higher compared to those from Qatar (60.5%) [18], Kuwait (53.1%) [11], Jordan (34.9%) [15], and Egypt (6.0%) [16]. Our results were consistent with an Indonesian study, where most of their participants (93.3%) were intended to be vaccinated against COVID-19 [25]. 

Possible explanations for the acceptance of participants in our study to be vaccinated against COVID-19 are the ongoing awareness efforts and high confidence in the government and the healthcare system. Our results support these explanations, as 82.2% of our participants exhibited high trust in the COVID-19 vaccines utilized in Saudi Arabia. Highly reported determinants of this trust include confidence in the government authorities (96.5%) and the healthcare system (94.2%). The government authorities contributed significantly to efforts that combated COVID-19 and ensured the safety of the vaccines. For example, the Saudi Food and Drug Authority was responsible for assessing and evaluating the safety, efficacy, and quality of COVID-19 vaccines before their utilization in Saudi Arabia [23,26]. In support of this, Al-Mohaithef and Padhi [13] found that those who had trust in the healthcare system were almost three times more intent to receive the COVID-19 vaccine than their counterparts. On the other hand, a lack of trust in government agencies was identified in another study as a predictor of COVID-19 vaccine hesitancy [17]. It is also possible that one of the reasons for the high rate of COVID-19 vaccine acceptability in our study, is that most of those who refused vaccination did not respond to the survey.

The strongest motivation that drove our study participants to receive the COVID-19 vaccine was their willingness to protect themselves and others from the disease (95.8%). Previous studies found that participants who had concerns about the risks of contracting COVID-19 exhibited a greater intent to receive the vaccine [13,23]. One study revealed that participants who perceived the risk of becoming infected with COVID-19 were 2.13 times more likely to be vaccinated than those who showed a lesser concern [13]. Another essential motivator as indicated by participants who were willing to accept the COVID-19 vaccine was the high confidence in the science and safety of the vaccines (84.3%). In the open-ended section of the questionnaire, one of our participants wrote, “no vaccine will be approved by the scientific authorities to be provided to millions [of people] unless its safety was confirmed.” More available information on the vaccines’ safety and effectiveness also contributed to promoting the acceptance of COVID-19 vaccines [11]. In support of this, more than three-quarters of our study’s participants reported the availability of awareness information on vaccines as the reason for their vaccination acceptance. Influence by the media (51.8%) and by other individuals such as community figures (67.2%), family members (54.5%), and friends (43.5%) were reported by participants of this study to be additional motivators to receive COVID-19 vaccines. Individuals are influenced by people around them as well as media including social media platforms, according to Jackson et al. [27] “A change in one partner’s health behavior is often associated with a change in the other partner’s behavior” (p. 385). Thus, it is important to continue health education campaigns that utilize social media platforms to form an awareness in the public community regarding COVID-19 vaccines, where people can influence each other positively. Interestingly, a few participants believed that vaccination would become compulsory in the future. For instance, one participant wrote in the open-ended space that “I want to take the vaccine now because it will be compulsory for everyone in the future.” In the same way, prior studies from Saudi Arabia found that making the vaccination compulsory was a significant predictor of willingness to receive the COVID-19 vaccine [23,24].

A few participants of our study (6.1%) refused to receive the COIVD-19 vaccine. Of these participants, 92.0% were worried about the side effects of the vaccine, 80.0% were concerned about the expedited vaccine trials, and 64.0% generally lacked trust in the vaccine. Similar factors were reported in prior studies [11,23,24]. Concerns regarding the vaccine’s safety and effectiveness, and fears regarding the potential side effects were the most critical determinants of vaccination refusal [12,23]. Alqudeimat et al. found that participants who believed that the vaccines introduced health-related risks were less willing to accept vaccination [11]. Another study from Saudi Arabia revealed that the vaccine’s efficacy and safety were among the most crucial factors that contributed to vaccine hesitancy [24]. Although many people were worried about the possibility of unknown long-term side effects, the Center for Disease Control and Prevention (CDC) in the United States confirmed that long-term side effects were unlikely [28]. Side effects have historically occurred within 6 weeks of receiving a vaccine dose. Thus, the United States Food and Drug Administration mandated that each COVID-19 vaccine recipient be examined for at least 2 months after the final dose [28]. 

Other factors reported by our participants for refusing COVID-19 vaccination include the beliefs that a vaccine is an unnecessary procedure (64.0%), that preventive measures are enough for combating the disease (52.0%), and the influence of the anti-vaccine movement (56.0%). The anti-vaccine movement on social media platforms influenced many people worldwide to accept or refuse the COVID-19 vaccination [29,30]. A Saudi cross-sectional study found that one-third (33.1%) of their participants were influenced by the anti-vaccine movement [24]. The spread of misinformation by the anti-vaccine movement through social media platforms intensified doubts about the vaccine among the general public and, in turn, decreased vaccine acceptability [23]. 

Using a chi-square test and the logistic regression in our study, the acceptability of the COVID-19 vaccine was not associated with the factors of gender, age, marital status, the discipline of study, education level, whether the participant contracted COVID-19, whether a family member or a friend contracted COVID-19, or whether a family member or a friend died from COVID-19. This finding contradicts several previous studies where demographic characteristics such as gender, age, and marital status were statistically correlated with the acceptability of the COVID-19 vaccine [11,13,23]. Moreover, in contrast to our study, previous studies found that contracting COVID-19 was associated with a strong intention to receive the vaccine [23,24]. A possible explanation for these discrepancies is that our study sample was mostly young (21.62 years), not-married (92.1%), and only 11.1% had contracted the disease.

Interestingly, we found significant associations between regularly receiving the flu vaccine and the acceptability of the COVID-19 vaccine. This result was supported by findings from prior studies [11,12,23]. Thus, it seems that pre-existing attitudes toward vaccines in general influence the acceptance of new vaccines. For example, an Australian study found that participants who had accepted the flu vaccine were five times more likely to accept the H1N1 (swine flu) vaccine [31]. Likewise, a study from Kuwait found that participants who regularly received flu vaccines were more intent to accept the COVID-19 vaccine [24]. 

We would like to acknowledge several limitations of this study. A key limitation is that the snowball (nonrandom) sampling technique was used due to circumstances associated with the COVID-19 pandemic and the social distancing procedures. The recruitment process was not under control of the team conducting the study, but controlled by people addressed in the first round of the approach. However, we distributed the questionnaire through social media platforms and through internal student groups to reach most university students. Another limitation is the possible bias in the data where people who refuse vaccination may have not responded to the questionnaire. Further limitation was that the data were collected using the Arabic version of the questionnaire. Thus, this study did not target expatriate students from non-Arabic speaking backgrounds. Finally, all participants came from Jazan University, so our results may not be generalizable to other Saudi Arabian Universities. Despite these limitations, the current study contributes significantly to COVID-19 research and provides insight into the acceptability of the COVID-19 vaccine and its determinants among Jazan University students in Jazan, Saudi Arabia.

## 5. Summary, Conclusions, and Perspective

In total, 90.4% of participants who were not vaccinated against COVID-19 stated they would receive the vaccine if given the opportunity by health authorities. More than three-quarters of the participants declared their trust in the COVID-19 vaccines used in Saudi Arabia. The primary factor for the participants’ trust in the COVID-19 vaccine used in Saudi Arabia was their confidence in the government and the healthcare system. The main motivators of a participants’ acceptance to receive the vaccine include preventative purposes, confidence in the vaccine’s safety, public awareness information regarding the vaccine, and influence from the media and people such as family members, friends, and community figures. Although most of the participants had high vaccine acceptability, a small group refused to receive the vaccine due to concerns regarding side effects, expedited vaccine trials, and a general lack of trust in the vaccine. Continuing to enhance people’s confidence in the efficacy and safety of the vaccine through public awareness is integral to the success of vaccination efforts. It may be appropriate for universities to urge reluctant students to receive the vaccine voluntarily by launching peer-support programs.

Generally, the findings of this study will help researchers understand the current situation, especially as Saudi universities plan to apply in-person education from the beginning of 2022. Almost all educational institutions applied distance education during the previous period of the pandemic. This study will also bridge the current gap in local research related to the acceptability of the COVID-19 vaccine among university students.

In terms of further research, it would be interesting to follow up with participants of this study who were not vaccinated at the time of data collection to investigate if they received the vaccine since then and their reasons for doing so. Such data would help clarify changes in people’s intentions toward novel vaccines over time and identify related determinants. In addition, future research should consider students from non-Arabic speaking backgrounds.

## Figures and Tables

**Figure 1 vaccines-09-00943-f001:**
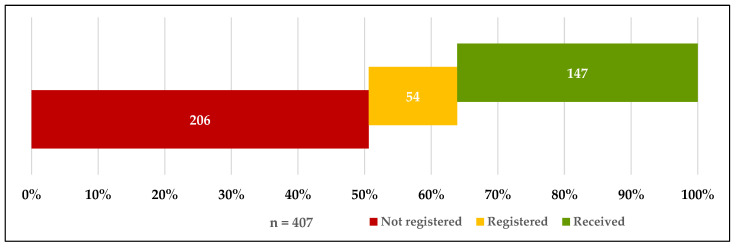
Participants who had received, were registered or were not registered for the COVID-19 vaccine.

**Figure 2 vaccines-09-00943-f002:**
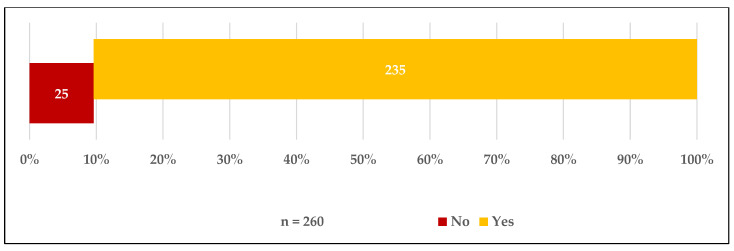
Acceptance to receive the COVID-19 vaccine.

**Figure 3 vaccines-09-00943-f003:**
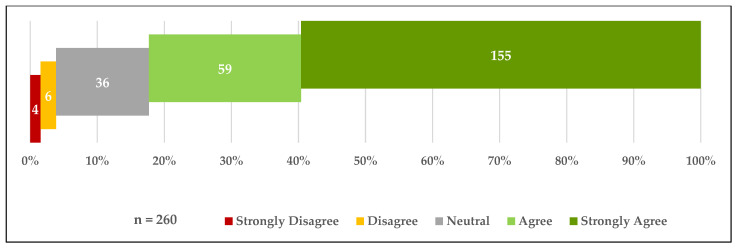
Trust toward COVID-19 vaccines provided in Saudi Arabia.

**Figure 4 vaccines-09-00943-f004:**
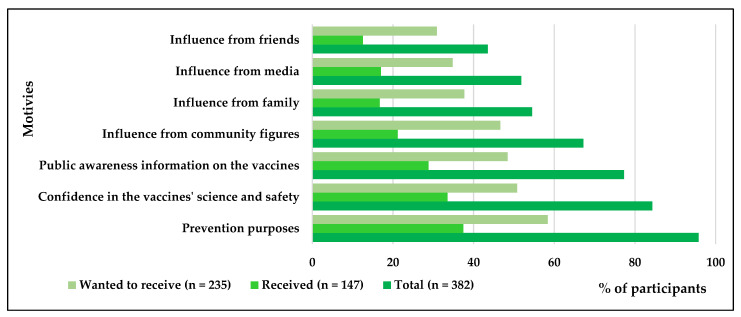
Motives of participants who received or wanted to receive COVID-19 vaccines.

**Figure 5 vaccines-09-00943-f005:**
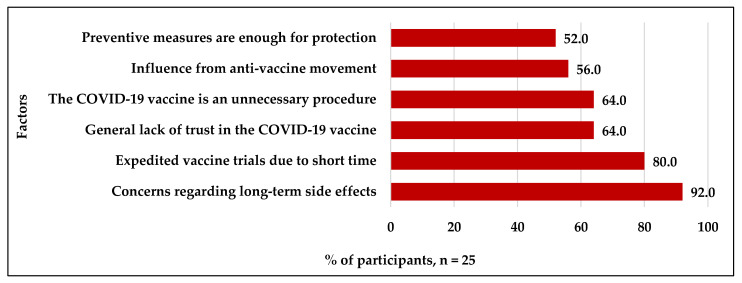
Factors that drive the refusal of COVID-19 vaccines.

**Table 1 vaccines-09-00943-t001:** Demographic information.

Variables (*N* = 407)		*n*	%
**Gender**	Male	227	55.8
	Female	180	44.2
**Age**	≤22 years	296	72.7
	>22 years	111	27.3
**Marital status**	Not-married	375	92.1
	Married	32	7.9
**Discipline of study**	Non-health related	180	44.2
	Health-related	227	55.8
**Education level**	First academic year	104	25.6
	Middle academic year	150	36.9
	Final academic year	153	37.6
**Contracted COVID-19**	No	362	88.9
	Yes	45	11.1
**A family member or a friend contracted COVID-19**	No	76	18.7
	Yes	331	81.3
**A family member or a friend died due to COVID-19**	No	325	79.9
	Yes	82	20.1
**Regularly received the flu vaccine**	No	211	51.8
	Yes	196	48.2

Note: In the education level: First academic year = academic levels 1 and 2, middle academic years = academic levels 3–6, final academic year = academic levels 7 and 8.

**Table 2 vaccines-09-00943-t002:** Factors influencing trust toward the COVID-19 vaccines provided in Saudi Arabia.

Factors (*N* = 260)	Agree	Neutral	Disagree
	*n* *(%)*	*n* *(%)*	*n* *(%)*
Confidence in the government	251 (96.5)	8 (3.1)	1 (0.4)
Confidence in the healthcare system	245 (94.2)	14 (5.4)	1 (0.4)
Important community figures who received the vaccine	223 (85.8)	28 (10.8)	9 (3.5)
Information about vaccines	212 (81.5)	41 (15.8)	7 (27)
WHO public advice on vaccines	210 (80.8)	39 (15.0)	11 (4.2)
Experience and reputation of manufacturers	205 (78.8)	49 (18.8)	6 (23)

Note: *N* = total number of participants who responded to this question; answer options = Agree (Agree and Agree Strongly), Neutral, and Disagree (Disagree and Strongly Disagree); *n* = number of responses to each item; *%* = percentage of the responses to each item.

**Table 3 vaccines-09-00943-t003:** A chi-square test of independence between the acceptability of the COVID-19 vaccine and the regularly received the flu vaccine variable.

Variable (*N* = 407)	*n*	X^2 a^	df	sig.
Regularly received the flu vaccine		7.300	1	0.004
No	211 (51.8%)			
Yes	196 (48.2%)			

Note: The dependent variable is the acceptability of the COVID-19 vaccine (0 = hesitancy, 1 = acceptability). All tested independent demographic variables were identified in the data analysis section. ^a^ *X*^2^ = chi-square (for 2 × 2 tables, continuity correction was computed, while for larger tables, Pearson chi-square was computed). *df* = degrees of freedom, sig. = *p*-value.

**Table 4 vaccines-09-00943-t004:** Binary logistic regression analysis on factors significantly associated with the acceptability of the COVID-19 vaccine.

Independent Variable		B	Std. Error	Wald	df	Sig.	Odd Ratio	95% CI	
								Lower	Upper
Regularly Received the Flu Vaccine									
	0 = No	−1.396	0.520	7.204	1	0.007 *	0.248	0.089	0.686
	1 = Yes	0 ^a^			0				
Method: Enter.									

^a^. This parameter is set to zero since it is redundant. Dependent variable is “the acceptability of the COVID-19 vaccine”. Only significant coefficient is presented in this table—“regularly received the flu vaccine”. * *p* < 0.05. All tested independent variables were identified in the data analysis section.

## Data Availability

The data presented in this study are available on request from the corresponding author.

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
