# Peer review of "Acceptability of the COVID-19 Vaccine and Its Determinants among University Students in Saudi Arabia: A Cross-Sectional Study"

_vaccines, 2021, doi:10.3390/vaccines9090943_

Round 1

Reviewer 1 Report

Dear Authors,

The research paper titled "Acceptability of the COVID-19 vaccine and its determinants among university students in Saudi Arabia: a cross-sectional study" is very well written and logically constructed as well.

I have just a few minor concerns about the results interpretation and few suggestions in order to improve the quality of your manuscript.

1- Figure 3: the square symbol of the graph legend about "Strongly disagree" should be in red color and not in orange one.

2- Figure 4: it should be splitted in two panels or two figures. One concerning people that received the vaccine, the other one concerning the people that are willing to receive the COVID-19 vaccine.

3- Concerning the results about the unwillingness of COVID-19 vaccination, in my opinion the should be associated with other parameters. For example, the number of participants that "reported an unwillingness to receive the COVID-19 vaccine" have a non-health related study background or not? Have they contracted the COVID-19 or a member of their family contracted it?

4- Furthermore, concerning the vaccinated people, a similar correlation analysis should be performed in order to reinforce the paper conclusions.

Author Response

Corrected. See figure 3.

The data in the figure was split to three bars: received COVID-19 vaccines, wanted to receive COVID-19 vaccines, and total (Figure 4).

Done (lines: 180-184, 268-279).

Done (lines: 180-184, 268-279).

Reviewer 2 Report

This manuscript present data on attitude towards vaccination for coronavirus in a population of students at Jazan University, Saudi Arabia. The authors developed a questionnaire in english and in arab language (included in the supplementary information), and conducted a prestudy statistical evaluation to determine the number of participants needed. The actual number included in the analysis (n=407) exceeded this number needed (n=384). Parameters included demographic data, presence of COVID-19 in friends or family, being vaccinated with flu-vaccine, various aspects of acceptance and registration for vaccination. The outcome of this study showed that most participants were willing to receive the vaccine, and that confidence in the government and in the health system were  most relevant in a positive attitude to vaccination, and that prevention purposes was a main basis for motivation.

This is an interesting study, albeit it contains probably some bias created by the design (“snowball technique”). The presentation of the design, conduct of the study, and data collection and interpretation is clear. The Introduction and Discussion can easily be reduced.

There are a number of suggestions to improve the quality of this manuscript:

  • A main aspect in a proper understanding of the data, both in the literature overview and in the present study, is the wording of various attitudes towards vaccination. The only term which is properly defined is the word “hesistancy”(lines 65-66), but it is not clear whether the use of this word and its derivatives (like “vaccine-hesitation”) throughout the manuscript is properly compliant with this definition. This aside, a variety of wordings is used, such as “unwilling”, “refuse”, “vaccine-resistant”, “hesitant”, “not sure” or “unsure”, “not willing to receive” and more. It is advised not to use multiple wordings for the same or similar attitudes, and use selected words consistently. It is understandable that different words for the same attitude are used in literature, but readability in case of using only selected descriptors is strongly increased. I guess that major categories are “acceptance”, “hesitation” (or “reluctance” although this is a stronger term) and “refusal”.
  • Second, reasons for hesitancy are given in lines 107-112, but not for the other descriptors/categories describing attitudes. It can be assumed that similar reasons apply in principle for “hesitation” and “refusal”. But the relative value of distinct reasons may be variable: for instance, people believing conspiracy theories might more easily be inclined to refuse vaccination than be hesitant. Interestingly, “influence from anti-vaccine movement” is not a major factor in refusal of vaccination (Figure 5), which may need some discussion. Reverse, trust in government is more important for hesitation than in refusal. Noteworthy, different attitudes towards vaccination directly influences approaches to attract people to accept vaccination. It is strongly recommended to give this more attention in the text, so that the data can be better interpreted. For instance, Table 2 and other data indicate that most participants were in the category “hesitant”. See also Figures 2 and 4. This interpretation affects also the conclusions of the manuscript, since most participants were willing to be vaccinated or already vaccinated.
  • Following the point above, and taking in consideration the design of the study, it cannot be excluded that there is bias in the data towards “hesitation”; in other words, people who refuse vaccination have not responded to the questionnaire. This needs to be mentioned in the paragraph about limitations of the study at the end of the Discussion. A possible bias in the data towards “hesitation” can also affect the explanation given in lines 311-319 of the Discussion, which may need some adaptation.
  • Since the number of participants refusing vaccination in this study is rather low (n=25), a detailed discussion like in lines 347-370 might not be warranted.
  • The Introduction and the Discussion are by far too large-sized. The literature overviews in these sections are interesting, but provide too much information than is necessary to underline the aim of the study, such in the context of what is already in the literature. It is proposed to summarize the relevant literature in tabular format, which will shorten the Introduction substantially. It is proposed to shorten the Discussion so that there is a prime focus on the interpretation of the Results. Literature can be cited so that the data of his study are in a proper context.
  • At the end of the Introduction the aim of the study is presented, line 114-116. The subsequent sentences are not part of the aim, but rather put the study in a perspective. It is suggested to move these sentences (line 116-120) to the end of the Discussion or to the Conclusions. It is advised to give more detail on general aspects related to the aim, for instance describe the “related determinants” (line 116) in more detail, including a rationale for selection of determinants.
  • The Conclusions at the end of the manuscript (line 400-420) are for the most part not conclusions, but rather present a summary of findings and a perspective. This can be in this section, but then the section heading should then be changed in “Summary, Conclusions and Perspective”. Alternatively, perspectives can be given in a separate paragraph at the end of the Discussion, like the paragraph on limitations (line389-399). This aside, the conclusions need to phrased as concluding statements, like “The primary factor … healthcare system” (line404-405).
  • The authors mention that a “snowball technique” was used in recruitment of participants, already in the abstract. This is further explained in section 2.3 of the Materials and Methods. It is then clear that the “snowball technique” is not a standard tool in survey studies, and this is well explained in the text. It is also correct to describe this as a study limitation, because the recruitment process is not under control of the team conducting the study, but controlled by people addressed in the first round of the approach. However, as a consequence it is questionable to describe the study as a cross-sectional study (line 124), and also not as a study in which all participants came from Jazan University (line 125, line 395-396) because the “snowball” might have left the environment of Jazan University. Note in this respect that the location of participants is not included in the questionnaire (supplementary material). It is strongly advised that this is described in general terms in the paragraph on the aim of the study (Introduction), and in more detail in the Material and Methods.
  • Table 3 lacks some data needed to understand the outcome of statistical analysis: acceptability should be presented for each of the input groups in statistical analysis. Also, “education level” needs clarification. Significance levels should be included in section 2.4, Data analysis, and in case of being not significant, values should not be given but rather “not significant” or “n.s.” should be mentioned.
  • The authors should consider a multivariate statistical evaluation to expand data in Table 3. This may require consultancy of a statistical expert.

Minor comments

  • It is assumed that the participants were not individually known to the team conducting this study, as this was not included in the questionnaire: this aspect of privacy should be mentioned in the text: this exceeds the request for consent (line 167).
  • Line 55: explain the word “Sehaty”.
  • Line 66: remove “(p.4163)”.
  • Line 84: “histant” seems to be a typographical error; you mean “hesitant” or “resistant”?
  • Line 92”: “Recent statistics by the WHO”: is this worldwide or in a region of the world? Please clarify.
  • Line 100-101: “…46.0% had vaccination hesitancy, 6.0% refused the vaccine, and only 6.0% were willing to be vaccinated [17].” What is the remaining 42%?
  • Line 220-221, Figure 1: change “unregistered” into “not registered”, because “unregistered” has a different interpretation.
  • Line 28: “scientific” is not appropriate and should be deleted.
  • Line 295: delete “high” in “… the results showed that the high acceptability …”.

Author Response

1. We use the following words consistently throughout the manuscript as suggested by the reviewer: acceptance, hesitation and refusal, and deleted all other terms.

We also defined the three words: acceptance, hesitation and refusal for the purpose of this study (lines 72-76).

2. We believe that the influence from anti-vaccine movement is one of the top factors for vaccine refusal in this study (56.0%) (Figure 5) (Lines: 360-367).

We believe that the indicated information by the reviewer regarding "trust in government" is related to the level of acceptance. It was discussed in details in lines: 301-313.

3. The possible bias was added to the paragraph of limitations (lines 393-395).

The following paragraph was added to the discussion section: "It is also possible that one of the reasons for the high rate of COVID-19 vaccine acceptability in our study, is that most of those who refused vaccination did not respond to the survey" (lines 314-316).

4. The idea is that there are factors for refusal of vaccine regardless of the number of participants refusing vaccination in this study. According to the reviewer's suggestion, the sample is not random so a bias of data is possible, and people who refuse vaccination may not have responded to the questionnaire. We believe it is important to discuss the factors of refusal in detail. If you insist the reduction of this part, we will follow your instructions.

5. The introduction and the discussion were shorten according to the reviewer's suggestions.

6. The indicated sentences were altered and moved to the conclusions as recommended by the reviewer (lines 417-421). 

7. The heading changed to summary, conclusion and perspective (line 402). The conclusions were phrased as concluding statements

8. According to the suggestion of the author, snowball technique was described as a study limitation (lines 387-391).

It was clearly indicated in the information sheet and in the link distribution message that the population of the study are the students of Jazan University. In addition, this was clarified in the manuscript (lines 157, 164-167) as suggested by the reviewer.

Regarding the cross-sectional design, the data were collected at a single point of time (April 2 and April 23, 2021), despite sharing the link of survey among respondents. Many peer-reviewed published cross-sectional studies* were conducted using the snowball technique. If you believe that the sampling technique we used is not snowball, we welcome any suggestions on this regard.

Otherwise we can remove cross-sectional study from the title and use "an online survey study".

* For example:

Alfageeh, E.I.; Alshareef, N.; Angawi, K.; Alhazmi, F.; Chirwa, G.C. Acceptability of a COVID-19 vaccine among the Saudi population. Vaccines 2021, 9, 226.

9.The significant results were presented in the table. Non-significance of other variables was stated in the text. Education level was clarified. Other information was added to the table and its notes (lines 257-266).

Significance level was added to the section 2.4 (lines 184-185).

10. Multivariate statistical evaluation were conducted using logistic regression (lines: 180-184, 268-279).

11. Added according to the reviewer's suggestions (lines: 162-167).

12. Explanation of "Sehaty" was added (lines: 52-56).

13. Removed (line: 66).

14. Corrected (line: 82).

15. In Saudi Arabia (line: 91).

16. More details were added to the text (lines: 100-101).

17. Changed to "not registered" (lines: 208-209).

18. Deleted (line: 221, table 2).

19. Deleted (line: 294).

Round 2

Reviewer 2 Report

THe authors have addressed the points raised about the original submission, and made some revisions in the manuscript.